# Novel Molecular Classification of Breast Cancer with PET Imaging

**DOI:** 10.3390/medicina60122099

**Published:** 2024-12-21

**Authors:** Ngô Minh Toàn

**Affiliations:** 1Gyula Petrányi Doctoral School of Clinical Immunology and Allergology, Faculty of Medicine, University of Debrecen, H-4032 Debrecen, Hungary; ngo@mailbox.unideb.hu; 2Medical Imaging Clinic, Clinical Centre, University of Debrecen, H-4032 Debrecen, Hungary

**Keywords:** breast cancer, positron emission tomography, PET biomarkers

## Abstract

Breast cancer is a heterogeneous disease characterized by a wide range of biomarker expressions, resulting in varied progression, behavior, and prognosis. While traditional biopsy-based molecular classification is the gold standard, it is invasive and limited in capturing tumor heterogeneity, especially in deep or metastatic lesions. Molecular imaging, particularly positron emission tomography (PET) imaging, offering a non-invasive alternative, potentially plays a crucial role in the classification and management of breast cancer by providing detailed information about tumor location, heterogeneity, and progression. This narrative review, which focuses on both clinical patients and preclinical studies, explores the latest advancements in PET imaging for breast cancer, emphasizing the development of new tracers targeting hormone receptors such as the estrogen alpha receptor, progesterone receptor, androgen receptor, estrogen beta receptor, as well as the ErbB family of receptors, VEGF/VEGFR, PARP1, PD-L1, and markers for indirectly assessing Ki-67. These innovative radiopharmaceuticals have the potential to guide personalized treatment approaches based on the unique tumor profiles of individual patients. Additionally, they may improve the assessment of treatment efficacy, ultimately leading to better outcomes for those diagnosed with breast cancer.

## 1. Introduction

Breast cancer remains a significant global health concern, with about 2.2 million new cases and 685,000 deaths reported in 2020 [1]. The risk of developing breast cancer escalates with age, particularly affecting individuals between the ages of 60 and 69 [2]. Despite improving trends in breast cancer mortality rates, there has been a rapid increase in incidence, particularly in developed countries [1].

To effectively address the increasing burden of breast cancer, it is crucial to investigate the disease at a molecular level. Breast cancer is classified into four molecular subtypes: Luminal A, Luminal B, HER2-enriched, and Triple-Negative Breast Cancer (TNBC). This classification is based on the positivity of receptors (estrogen, progesterone, and HER2 receptors) and the level of Ki67 [3]. Each subtype has distinct risk factors [4], prevalence patterns [2,5], and treatment plans. In addition to variances in surgical and radiation approaches, the chemotherapeutic treatment of each breast cancer subtype also differs; hormone-positive subtypes typically respond to endocrine therapy, HER2-positive subtypes require HER2-targeted therapy, and TNBC relies on cytotoxic agents [3]. These differences may lead to different adverse effect profiles of the chemotherapy and differences in medical expenses [6,7]. Furthermore, the behavior and prognosis of each subtype vary. Luminal A tumors generally have the highest survival rates, whereas TNBC is characterized by aggressiveness and a greater tendency to metastasize to the brain and lungs, unlike other subtypes, which may also metastasize to the liver [8,9].

Traditionally, molecular classification relied on biopsy-based samples using methods like immunohistochemistry (IHC) or fluorescent in situ hybridization (FISH). However, these approaches are invasive, uncomfortable, and cannot effectively access deep or metastatic lesions. Cancer cells undergo genetic mutations and cellular changes that drive progression, adaptation, and drug resistance. These mutations accumulate, creating genetic diversity and tumor heterogeneity [10]. Breast cancer exhibits significant levels of tumor heterogeneity, influencing therapy sensitivity and patient overall survival [11]. Even though it is considered the gold standard, tissue-based sampling has intrinsic limitations in accessing both intra-tumor and inter-tumor heterogeneity, particularly considering the complexity and progression of breast neoplasms [12].

Imaging methods have been implemented to classify the breast cancer subtypes using different patterns observed in many ultrasound, MRI, and CT techniques [13,14,15,16,17]. Moreover, [^18^F]Fluorodeoxyglucose (FDG) studies have revealed significant metabolic variations between luminal and non-luminal subtypes [18]. Additionally, studies implemented angiogenesis Arginylglycylaspartic acid (RGD) imaging to classify the subtypes better, using a metabolic activity-to-angiogenesis ratio [19,20]. These approaches are based on the distinct behaviors observed across the subtypes: luminal subtypes exhibit a higher rate of desmoplastic reaction, while non-luminal subtypes, particularly TNBC, are more aggressive, with increased metabolism and frequent central necrosis. The HER2-enriched subtype shows heightened angiogenesis. However, these imaging techniques are nonspecific and cannot accurately determine tumor receptor status and molecular subtypes with high confidence.

With recent attention and advancements, Positron Emission Tomography (PET) molecular imaging has emerged as a promising tool for assessing breast cancer biomarkers thanks to its high quantitative ability and resolution [21]. Furthermore, numerous additional biomarkers (EGFR, HER3, VEGF, androgen receptors, etc.) have been identified as playing significant roles in treatment options and prognosis, supplementing the conventional classifications.

Molecular imaging of breast cancer enables whole-body, non-invasive investigation of receptor statuses and other molecular biomarkers using novel radiopharmaceuticals. These biomarkers can change, particularly during metastasis or after treatment, complicating prognosis. Molecular imaging can dynamically track these changes, providing critical insights into tumor heterogeneity, subtype-specific characteristics, and cancer behavior patterns, thus allowing the precise identification of subtypes and biomarkers, guiding timely targeted therapies that maximize efficacy while minimizing side effects.

This narrative review summarizes the latest advancements in molecular imaging for breast cancer classification, targeting both conventional (Table 1) and novel biomarkers (Table 2) in tumor and infiltrating immune cells (Figure 1). We specifically focus on the most recent radiotracers and their application from preclinical research to clinical settings. These advancements have the potential to optimize therapy selection and monitor therapy responses, ultimately improving personalized treatment strategies in the future.

## 2. Conventional Biomarkers Imaging

### 2.1. Human Epidermal Growth Factor Receptor 2 Imaging

Human epidermal growth factor receptor 2 is one of the receptors in the epidermal growth factor family, encoded by the erbB-2 oncogene. Its protein overexpression or gene amplification has been present in around one-fifth of breast cancer cases [22], especially in HER2-enriched or HER2+ luminal B subtypes, which have a worse prognosis than their HER2-negative counterpart [23]. Upon dimerization, the receptor initiates tyrosine kinase activity and a cascade of events that regulate cell proliferation and angiogenesis, promoting cancer invasion and metastasis [22]. Consequently, HER2-positive breast cancer has a higher risk of metastasis, with a predilection to spread to the bone, liver, and brain [24]. Even though many HER2 targeting therapies have shown promising results, HER2 breast cancer heterogeneity is a significant challenge in treating breast cancer patients. There is around a 28% conversion rate in HER2 status between primary tumors and subsequent recurrent or metastatic lesions, representing inter-tumor heterogeneity [25]. Furthermore, HER2 intra-tumor heterogeneity also plays a crucial role in therapy resistance and poor prognosis [26,27]. Recently, the HER2-low subtype (IHC +1 or IHC +2 with negative FISH) has been classified, requiring a distinct approach, with first-line therapy relying on trastuzumab deruxtecan and necessitating a dedicated assessment of HER2 levels [28]. Therefore, HER2 imaging offers a non-invasive method to assess the HER2 status of both primary tumors and metastases, as well as heterogeneity status, thus helping in therapy selection, predicting, and following up on therapy response.

#### 2.1.1. Trastuzumab Labeled with ^89^Zr

The use of the radiolabeled isotope ^89^Zr, with an extended half-life of approximately 78 h, is effective for labeling antibodies (also with long biological half-life). Trastuzumab labeled with ^89^Zr using the chelator DFO has been extensively evaluated in clinical settings. In most trials, it is common practice to administer a “cold” dose of trastuzumab, typically around 50 mg, prior to the tracer injection to reduce background signal. Scanning is usually performed 4 to 5 days after tracer administration [29,30]. However, studies suggest a lower cold dose may be more appropriate for patients already undergoing trastuzumab treatment [31]. The [^89^Zr]Zr-DFO-trastuzumab tracer has demonstrated its ability to detect or monitor HER2 status in metastatic lesions in the liver, bone, and lymph nodes regardless of the HER2 status of the primary tumor [29,30,31]. Notably, it can also detect brain metastases, likely due to tracer uptake facilitated by the disruption of the blood-brain barrier [32].

Despite these promising results, there is a recognized risk of discordance between the tracer uptake and IHC results, particularly in bone, liver, and lymph node lesions [33]. This discrepancy may lead to false positives, primarily attributed to the instability of DFO as a chelator for ^89^Zr. The free zirconium that is released tends to accumulate in bone tissue. Moreover, Fc-mediated uptake in active lymph nodes and the liver—key sites of antibody metabolism—can also result in false positives.

The tracer’s potential in following up therapy responses was demonstrated by Linders et al., using different parameters and ΔSUV_R_ (the difference between SUV ratios before and after therapy). However, the study included a small number of patients (n = 6) and reported some false-negative cases (two cases) likely due to HER2 downregulation following high-dose trastuzumab [34]. In larger clinical trials (e.g., the ZEPHIR trial), combining [^89^Zr]Zr-DFO-trastuzumab imaging with conventional FDG PET/CT significantly improved the ability to predict treatment response to trastuzumab emtansine (T-DM1), achieving positive predictive values (PPV) and negative predictive values (NPV) exceeding 90%, up to 100% [35,36]. Furthermore, a clinical trial by Gaykema et al. showed that [^89^Zr]Zr-DFO-trastuzumab imaging can help predict treatment responses when used alongside heat shock protein 90 (HSP90) inhibitors in patients. The results showed significant correlations between the ΔSUV_max_ and decrease in tumor size (*r*^2^ = 0.69) [37]. Additionally, in a preclinical study by McKnight et al. involving breast cancer tumor-bearing mice, the tracer was able to assess the effectiveness of tyrosine kinase inhibitors. There were significant correlations between tumor uptake and decreases in HER2 expression (*r* around 0.6) and in tumor volume (*r* around 0.8) [38]. These findings underscore the potential of [^89^Zr]-DFO-trastuzumab not only in monitoring HER2 status but also in evaluating the efficacy of different therapeutic approaches. Nevertheless, larger studies with standardized assessment methods are required, whether using the SUV_max_ lesion-to-background ratio, blood, liver, or contralateral side.

To address the in vivo instability of DFO, alternative chelators for [^89^Zr]Zr-DFO-trastuzumab have been developed and tested in preclinical models. For example, Deri et al. showed that HOPO improved stability with lower bone uptake but presented challenges in synthesis and lower tumor uptake [39]. DFO* (DFOstar) was demonstrated by Chomet et al. to significantly improve thermodynamic stability while maintaining similar tumor-to-background ratios (TBRs). This resulted in lower bone uptake and greater accuracy in detecting breast cancer bone metastases [40].

Site-specific conjugation produces a more homogeneous tracer, leading to more predictable specificity and pharmacokinetics. In a study by Vivier et al. involving humanized mice with breast cancer, both fully (by PNGaseF) and partially (by EndoS) deglycosylated antibodies were conjugated to DFO specifically at the heavy chain glycan site ([^89^Zr]Zr-DFO-^ss^trastuzumab-EndoS) or non-specifically ([^89^Zr]Zr-DFO-^nss^trastuzumab-PNGaseF). Both strategies showed significantly higher tumor uptake and an improved tumor contrast with significantly lower liver and spleen uptake compared to [^89^Zr]Zr-DFO-trastuzumab [41]. This improvement is attributed to the attenuation of FcγR interactions while maintaining high HER2 binding affinity, requires additional but facile enzymatic deglycosylation step.

#### 2.1.2. Trastuzumab Labeled with ^64^Cu

In addition to [^89^Zr]Zr-DFO-trastuzumab, the use of [^64^Cu]Cu-DOTA-trastuzumab has also been investigated in patients due to its comparable long half-life of 12.7 h and better resolution due to lower positron energy (0.656 MeV vs. 0.897 MeV) [42]. Studies suggest that administering a cold dose of 50 mg trastuzumab, followed by a 48 h incubation period would optimize the TBR [43]; however, the tracer’s tumor visualization was limited in patients undergoing trastuzumab therapy [44]. Like [^89^Zr]Zr-DFO-trastuzumab, [^64^Cu]Cu-DOTA-trastuzumab has proven effective in visualizing known brain metastases larger than 1 cm with high confidence and HER2 specificity (validated with IHC and autoradiography) [45]. There is some discordance between FDG PET imaging and [^64^Cu]Cu-DOTA-trastuzumab, with certain lesions showing positivity with one tracer but not the other [43], suggesting the potential supplementary role of the tracer to the conventional FDG.

Moreover, [^64^Cu]Cu-DOTA-trastuzumab has demonstrated potential in predicting therapy response in HER2-positive patients. Mortimer et al. showed that patients who responded to treatment exhibited significantly higher baseline uptake at 2 days post-injection compared to non-responders. Although the study’s small sample size (n = 10) limits the generalizability of these findings [46].

One challenge with ^64^Cu is its tendency to transchelate with serum compounds, highlighting the need for a more stable chelator. A preclinical study by Woo et al. showed that [^64^Cu]Cu-NOTA-trastuzumab offers high specificity (significant tumor uptake decrease when blocked in vivo with cold trastuzumab) and a lower uptake and absorbed dose to organs compared to DOTA, indirectly indicating greater stability [47]. A clinical pilot study using the same tracer by Lee et al. also reported a reduced organ dose and a higher uptake in HER2-positive lesions compared to negative ones [48]. Recently, the NODAGA chelator has emerged as a promising alternative; [^64^Cu]Cu-NODAGA-trastuzumab offers easier labeling and enhanced stability while retaining high specificity in vitro [49].

#### 2.1.3. Trastuzumab Labeled with ^52^Mn

With a similar extended half-life of 5.6 days, ^52^Mn, a promising candidate for antibody-based imaging, offers an attractive radiation profile compared to ^89^Zr and ^64^Cu. Its 29% positron emission with a low maximum energy (0.575 MeV) results in enhanced resolution [50]. Toàn et al. utilized BPPA, a bispyclen-based chelator, to label trastuzumab with ^52^Mn. The resulting tracer, [^52^Mn]Mn-BPPA-trastuzumab, demonstrated higher TBRs and effectively distinguished HER2-positive from HER2-negative tumors in preclinical models, outperforming [^52^Mn]Mn-DOTAGA-p-SCN-trastuzumab, though in vivo stability needs further enhancement [51]. Omweri et al. evaluated [^52^Mn]Mn-Oxo-DO3A-trastuzumab, which showed comparable specific activity to [^52^Mn]Mn-BPPA-trastuzumab but with improved stability. This tracer accurately evaluated HER2 expression, showing significantly higher tumor-to-muscle ratios in HER2-positive tumors compared to HER2-negative ones in preclinical models [52].

#### 2.1.4. Pertuzumab Labeled with ^89^Zr

Imaging the HER2 receptor with a trastuzumab-based tracer during trastuzumab therapy can be challenging due to the saturation of the receptor’s binding sites. An alternative approach using pertuzumab, which binds to a distinct epitope on HER2, offers a potential solution. In a preclinical study by Marquez et al. using breast cancer xenografts, the [^89^Zr]Zr-DFO-pertuzumab tracer demonstrated high specificity, with significant tracer uptake observed during trastuzumab therapy [53]. Building on these findings, a patient study (n = 6) by Ulaner et al. revealed that optimal imaging was achieved 5–8 days post-injection with a cold dose of pertuzumab administered beforehand to enhance the TBR. The approach resulted in a slightly higher absorbed dose compared to [^89^Zr]Zr-DFO-trastuzumab. Nevertheless, the tracer proved effective in investigating HER2 heterogeneity and was able to detect brain HER2 metastases in some patients [54].

Beyond differentiating HER2 positivity, this tracer has been explored in preclinical models for its ability to follow up and predict therapy response. For instance, after T-DM1 treatment, the tracer visualized changes in HER2-positive tumor size more effectively than FDG. Specifically, the tumor’s tracer uptake did not significantly change, which is hypothesized to correspond to the unchanged HER2 concentration post-therapy [55]. In contrast, a study by Kang et al. with HSP90 inhibitor therapy showed decreased HER2 expression and a significant reduction in tumor uptake after treatment [56]. Another study by Lu et al. demonstrated the tracer’s predictive ability following paclitaxel treatment, where tumors with higher tracer uptake responded better, exhibiting reduced size and decreased FDG uptake (*r* = −0.59), correlating with lower HER2 expression post-therapy [57]. These findings highlight the tracer’s potential not only for characterizing HER2 expression but also for predicting and monitoring dynamic changes in HER2 status.

Similar to [^89^Zr]Zr-DFO-trastuzumab, a site-specific version, [^89^Zr]Zr-^ss^DFO-pertuzumab, developed by Vivier et al., showed improved tumor-to-spleen and liver ratios in preclinical models [58]. In a clinical study (n = 6) by Yeh et al., the site-specific tracer exhibited a similar total absorbed dose compared to non-specific [^89^Zr]Zr-DFO-pertuzumab, but with the kidney identified as the critical organ instead of the liver. Nevertheless, the site-specific tracer provided better lesion visualization in some HER2-positive patients than the non-specific version [59].

#### 2.1.5. Trastuzumab Emtansine Labeled with ^89^Zr

Trastuzumab emtansine (T-DM1) is an antibody-drug conjugate effective for some patients with trastuzumab-resistant cancers, but many either fail to respond or develop resistance over time. To improve therapy selection, [^89^Zr]Zr-DFO-T-DM1 can be employed over [^89^Zr]Zr-DFO-trastuzumab. In preclinical breast cancer models with varying HER2 expression, Al-Saden et al. demonstrated that this tracer shows high HER2 specificity, comparable to [^89^Zr]Zr-DFO-trastuzumab, and is particularly effective at stratifying HER2 status at 4 days post-injection [60]. A subsequent study, also in preclinical models using tumor-to-blood ratios, showed that the tracer could classify HER2 expression levels, with very strong correlations with HER2 expression (*r^2^* = 0.94) and, importantly, with tumor response to T-DM1 [61].

#### 2.1.6. Labeled Trastuzumab Fragments

Antibody fragments lacking the Fc region could offer superior alternatives due to their faster pharmacokinetics compared to full-length antibodies. In a study by Suman et al. using F(ab’) and F(ab’)2 fragments labeled with ^68^Ga via the NOTA chelator, the fragments retained their affinity, and the labeling process was gentler than with the conventional DOTA chelator, which can damage heat-labile fragments. As expected when using antibody fragments, the F(ab’) was primarily eliminated through the kidneys, exhibiting a faster clearance rate and consequently achieving higher tumor-to-organ ratios than F(ab’)2, except for the tumor-to-kidney ratios [62].

A study by Moreau et al. on breast cancer tumor-bearing mice, investigating the optimal chelator for the ^64^Cu labeling of trastuzumab’s Fab fragment, found that MANOTA provided better stability and superior tumor-to-blood and tumor-to-liver ratios (around twofold higher) compared to DOTA, DOTAGA, and even the previously mentioned NODAGA, which already had shown some improvements [63].

Various methods for the site-specific labeling of trastuzumab fragments have been explored. For instance, a modified Fab with a mutated light chain containing methionine—a rare amino acid typically hidden within protein pockets—enables easy and specific conjugation, enhancing labeling precision. Implementing the techniques in preclinical models, Yue et al. demonstrated that [^68^Ga]Ga-DFO-M74 trastuzumab’s Fab demonstrated higher affinity and stability, leading to faster clearance from organs and higher tumor uptake, resulting in an approximately twofold higher tumor-to-background than the wild-type tracer [64].

However, their small size makes antibody fragments rapidly cleared from the body, which may limit tumor uptake. Strategies such as adding polyethylene glycol (PEG) can extend their half-life, although this requires chemical conjugation. An alternative approach, PASylation—the addition of Pro, Ala, and Ser sequences—has been tested in pilot mice models by Mendler et al. [65] and in a human study (n = 1) with HER2-positive breast cancer by Richter et al. [66]. This technique showed prolonged plasma retention and successfully visualized primary tumors and metastases.

#### 2.1.7. HER2 Nanobody

Nanobodies, which are heavy-chain antibodies found in camelids, are even smaller than antibody fragments. HER2-targeting nanobodies, such as 2Rs15d and 5F7, are particularly well-suited for imaging with ^18^F due to their complementary short half-lives.

Zhou et al. utilized these two nanobodies with the novel fluorine prosthetic TFPFN for HER2-positive breast cancer xenograft imaging. In vitro results showed that [^18^F]TFPFN-5F7 exhibited a better binding affinity and a higher internalization rate than [^18^F]TFPFN-2Rs15d. However, 2Rs15d binds to a different domain (domain I) than trastuzumab (domain IV) and pertuzumab (domain II), making it a suitable imaging agent for patients undergoing HER2-targeting antibody therapies, whereas 5F7 binds to domain IV. Nevertheless, in vivo, both tracers showed high specificity for HER2 tumors and lower kidney uptake compared to other available ^18^F prosthetics. This reduced the radiation dose to the kidneys and improved tumor contrast, achieving tumor visualization comparable to residualizing labels [67]. Due to the similarly short physical half-life, [^68^Ga]Ga-NOTA-2Rs15d was studied by Gondry et al. and Keyaerts et al. in breast cancer patients, demonstrating good reproducibility, safety, and effectiveness in detecting tumor heterogeneity [68,69]. In a preclinical study by Ducharme et al. involving breast cancer xenografts with varying HER2 expression levels, [^89^Zr]Zr-DFO-2Rs15d was used to investigate the biodistribution of the nanobody at later time points. The tracer demonstrated high specificity for HER2 and was only partially blocked by trastuzumab in vitro, likely due to the induced internalization after binding. However, while small tracers like nanobodies are primarily cleared through the kidneys, the consistent and prominent kidney uptake observed up to day 3 poses a challenge for dosimetry [70].

#### 2.1.8. HER2 Affibody

Affibody molecules, which are protein scaffolds based on the Z domain of Staphylococcus Protein A, have an even smaller molecular size than nanobodies (just a few kDa) and exhibit exceptionally high affinity for the HER2 receptor (in the pM KD range). ZHER_2:342_ (ABY-002) was the first affibody tested in humans; however, it showed high liver uptake, which could impede the detection of hepatic metastases and necessitate modifications for optimal labeling—potentially altering the tracer’s in vivo behavior [71]. To address this, ZHER_2:2395_ was developed with an added cysteine at the C-terminus, increasing labeling possibilities [72]. Further refinement led to ZHER_2:2891_, which includes amino acid substitutions near the C-terminus to improve stability and increase hydrophilicity while retaining high affinity for HER2 [73]; adding a cysteine residue along with a DOTA chelator produces ABY-025.

[^68^Ga]Ga-ABY-025 has been clinically tested, demonstrating a lower absorbed dose compared to FDG [74] and effectively differentiating HER2 positivity, particularly through the TBR (optimally tumor-to-spleen ratio) [75]. It was also able to detect HER2 heterogeneity [76], which is especially beneficial in HER2-low patients [77]. While the correlation with IHC was not significant [77], the tracer did show a significant correlation with metabolic response to HER2-targeted therapy [78].

To reduce the hepatic uptake of ZHER_2:342_ tracers, Xu et al. developed MZHER_2:342_ by adding a hydrophilic linker and labeling the affibody with different isotopes. In preclinical models, [^18^F]F-Al-MAL-NOTA-Cys-MZHER_2:342_ successfully demonstrated HER2 specificity and showed a significant correlation with HER2 IHC scores (*r*^2^ = 0.99). This labeling strategy also further decreased liver uptake, along with low bone uptake, indicating enhanced tracer stability; however, kidney uptake remained high [79]. Similarly, [^68^Ga]Ga-MAL-NOTA-Cys-MZHER_2:342_ was able to differentiate HER2 positivity in patients and showed lower liver uptake compared to ABY-025 [80]. [^89^Zr]Zr-DFO-MAL-NOTA-Cys-MZHER_2:342_ in tumor-bearing mice also exhibited HER2 specificity, with reduced liver and osseous uptake, further suggesting the tracer’s stability [81]. Consequently, this highlights the flexibility and reproducibility of labeling MZHER_2:342_ and its potential for future studies.

Affibody molecules represent a promising approach for HER2 breast cancer imaging, and ongoing developments aim to enhance their performance. Recently, ZHER_2:2891_ was modified to ABY-027 (including an albumin-binding domain) to reduce kidney uptake and tested in preclinical models with [^177^Lu]Lu-ABY-027 for targeted radionuclide therapy [82]. Additionally, the newer affibody ZHER_2:41071_ features improved stability and hydrophilicity and is currently being evaluated in clinical studies using SPECT imaging with the ^99m^Tc-ZHER_2:41071_ tracer [83].

#### 2.1.9. Other Scaffolds and Peptides

Similar to affibodies, ADAPT6 (ABD-Derived Affinity ProTein), a scaffold derived from the albumin binding domain (ABD) of streptococcal protein G, can target HER2 with high affinity and rapid clearance. A study using (HE)3DANS-ADAPT6-GSSC labeled with ^68^Ga by different chelators showed high affinity for HER2-positive breast and ovarian cancer cell lines in vitro. In vivo testing on ovarian cancer-bearing mice revealed that [^68^Ga]Ga-(HE)3DANS-ADAPT6-GSSC-NODAGA achieved the best TBR (compared to DOTA, DOTAGA, and NOTA variants), highlighting its potential as a promising small molecule for HER2 targeting [84].

Another promising candidate is DARPins (designed ankyrin repeat proteins), small peptides with a high affinity for HER2. [^89^Zr]Zr-DFO-G3-DARPin, produced by Fay et al. through sortase enzyme site-specific conjugation, despite the slow conjugation process, demonstrated high affinity and specificity with excellent tumor contrast in breast cancer xenograft preclinical models [85].

Protein scaffolds of bacterial origin, or to a lesser extent from camelids, may trigger immunogenic responses [86]. Small synthetic peptides are a promising class of candidates for HER2 imaging due to their ease of modification and reduced immunogenicity. However, these peptides can be degraded by enzymes in vivo, so strategies like D- or beta-amino acid substitution, cyclization, and PEGylation are often employed to enhance their stability [87]. Among the studied peptides, two are particularly notable: KCCYSL, which has a high affinity with a Kd of 295 nM and binds to a different site than trastuzumab, and LTVSPWY, which, while having a lower affinity, binds to the same site as trastuzumab [88]. A study by Biabani et al. using [^68^Ga]Ga-DOTA-(Ser)_3_-LTVSPWY to assess HER2-positive breast cancer xenografts in mice showed high tumor affinity, with a significant decrease in uptake when blocked both in vitro and in vivo with an unlabeled peptide. The tracer demonstrated serum and in vivo stability, with rapid renal clearance due to increased hydrophilicity from serine residues. Despite high initial renal activity, uptake decreased rapidly after 1 h, maintaining an adequate tumor-to-blood ratio of 1.7 at 2 h post-injection [89]. Another study by Ducharme et al. compared [^68^Ga]Ga-DOTA-PEG_2_-DTFPYLGWWNPNEYRY and [^68^Ga]Ga-DOTA-PEG_2_-GSGKCCYSL, using phosphoramidon to protect against degradation. Although both tracers demonstrated significant differences between HER2-positive and HER2-negative tumors, the differences were inconsistent across time points (1h versus 2h post-injection) and did not show significant uptake decreases in the in vivo PET blocking study. Nonetheless, [^68^Ga]Ga-DOTA-PEG_2_-DTFPYLGWWNPNEYRY appeared more promising, with higher tumor uptake at 1 h post-injection and greater stability in human serum [90].

In conclusion, small molecules (nanobodies, affibodies, etc.) targeting HER2 offer significant advantages, including high affinity and rapid clearance, leading to high tumor contrast and lower absorbed doses. However, they commonly suffer from high kidney uptake, prompting various strategies to address this issue, such as administering Gelofusine, adding linkers like PEG or PAS, using non-residualizing chelators or prosthetics, and improving tracer stability [91]. However, these modifications can potentially alter the peptide’s function and biodistribution, underscoring the need for further research to develop optimal HER2 imaging agents.

### 2.2. Estrogen Receptor Alpha Imaging

Estrogen receptor positivity is present in over 60% of breast cancer patients [92], primarily driven by the alpha estrogen receptor, which determines the luminal subtypes [93]. Heterogeneity of estrogen receptor expression is observed in over one-third of advanced estrogen receptor-positive cases [94]. Furthermore, endocrine therapy, commonly used in these patients, can build up cancer resistance through receptor loss [95], receptor gene mutations, or the activation of related pathways, necessitating combination therapies [96]. Therefore, PET imaging that targets the alpha estrogen receptor provides valuable insights into receptor status in both primary tumors and metastases, supporting timely therapy adjustments and monitoring of treatment response.

In ER-positive patients, in comparison with [^18^F]FDG, imaging using [^18^F]FES (fluoroestradiol) has demonstrated improved detection of metastases in bone lesions, whereas [^18^F]FDG is more effective for non-bone lesions. Thus, combining these tracers can enhance metastasis detection and improve staging and re-staging accuracy [97,98,99].

Furthermore, a study in ER-positive patients by Iqbal et al. showed that [^18^F]FES can effectively predict response to anti-estrogen therapy, independent of ESR1 mutation status [100]. This tracer tracks changes in ER expression throughout treatment, with ER levels decreasing during therapy and rising with disease progression, making [^18^F]FES a valuable tool for monitoring ER dynamics [100]. To optimize patient selection for anti-estrogenic therapy, He et al. identified the ΔSUV_max_ (changes in SUV_max_) ≥38.0% after four weeks as the most accurate predictor of therapy response, outperforming both baseline and 4-week SUV_max_ [101]. A study by You et al. in metastasis ER-positive patients receiving aromatase inhibitors showed that [^18^F]FES can detect ER changes in a timely manner, including ER-negative conversion, enabling early switches to chemotherapy and potentially improving survival outcomes. This was especially evident in primary tumors with low ER, where around 70% of metastatic tumors showed [^18^F]FES negativity [102].

The tracer enables the detection of ER changes as early as 7 days following anti-estrogen therapy with or without docetaxel, as demonstrated by Liu et al. in preclinical models. When combined with [^18^F]FDG, [^18^F]FES provides valuable information on the decrease in ER expression as an indicator of docetaxel responsiveness following combined therapy [103]. A study by Gennari et al. found that a baseline SUV_max_ cut-off < 2 helps identify endocrine-resistant patients who benefit more from chemotherapy, showing longer progressive-free survival (PFS) with chemotherapy (23 months) compared to endocrine therapy (12.4 months) [104].

In progressive ER-positive patients previously eligible for endocrine therapy, treatment with vorinostat (an HDAC inhibitor) combined with an aromatase inhibitor showed >6 months without progression in 4/10 patients (n = 10) who stuck with the protocol, possibly due to vorinostat’s ability to restore ER sensitivity. Baseline [^18^F]FES imaging predicted response in FES-positive individuals, although no changes in tracer uptake were observed after vorinostat therapy, as shown by Peterson et al. [105]. In another study by Liu et al. involving ER-positive patients starting palbociclib in combination with endocrine therapy, [^18^F]FES successfully predicted treatment response, with significantly longer PFS seen in FES-positive patients (23.6 months vs. 2.4 months) and in those with a low heterogeneity index (HI) [106].

Another alternative tracer targeting the ERα receptor in breast tumors is 4-Fluoro-11β-methoxy-16α-[^18^F]fluoroestradiol ([^18^F]4FMFES). In a preclinical study by Paquette et al. involving ER-positive and ER-knockdown tumors, [^18^F]4FMFES was able to differentiate between positive and negative ER tumors and demonstrated significantly higher tumor uptake and TBR compared to [^18^F]FES. However, it also shows notable increased uptake in the uterus and ovaries. ER specificity was confirmed by a marked reduction in tumor uptake following fulvestrant administration [107]. Despite its higher abdominal uptake compared to [^18^F]FES, this can be mitigated using loperamide and anticholinergic agents, which enhance tumor contrast by reducing abdominal uptake [108]. In a clinical study by Paquette et al. involving ER+ patients, [^18^F]4FMFES was proven to be more stable and could detect more lesions, while organ uptake was significantly lower, including a reduction in blood uptake by 75%. However, the primary elimination route remained the gastrointestinal tract and liver, leading to persistently high liver uptake [109]. Other notable SPECT tracers that hold great potential for ER assessment, including [^131^I]IPBA-EE [110] and [^131^I]EITE [111], are currently under preclinical investigation.

### 2.3. Progesterone Receptor Imaging

There are two Progesterone receptor (PR) isoforms, PRα and PRβ. Progesterone is known to influence early events in breast carcinogenesis, with its effect on proliferation mainly via PRβ [112]. However, PRα also plays an important role in the effectiveness of anti-progesterone agents contributing to poor prognosis in anti-estrogen therapy [112,113]. Although there is no routine tool to differentiate the two isoforms, both play significant roles in breast cancer progression. PR-negative luminal cancer represents more than 10% of luminal cancer [114] and has been associated with worse outcomes [115,116]; adding chemotherapy can improve the prognosis for these PR-negative luminal cancer patients [116]. Additionally, as PR expression increases under estrogen’s effect during endocrine therapy, it can serve as an indirect marker to predict and monitor therapy response by tracking receptor expression throughout treatment [117].

[^18^F]FFNP is a tracer that targets the PR for imaging. In breast cancer patients, Dehdashti et al. have shown that the tumor-to-non-tumor uptake ratio of [^18^F]FFNP can effectively differentiate PR positivity, with uptake ratios correlating to PR status [118]. However, similar to estrogen receptor tracers, the liver and gastrointestinal tract show high activity of [^18^F]FFNP, which may impede lesion detection in these areas [118]. Nevertheless, Chan et al. demonstrated on preclinical models of breast cancer that [^18^F]FFNP could detect tumor’s hormone receptor changes more sensitively and earlier (day 4) than [^18^F]FES or FDG. This allows earlier monitoring of ERα expression following estrogen deprivation therapy through a decrease in PR expression corresponding to decreased tumor uptake [119]. Conversely, low-dose estradiol-induced an increase in PR, which was trackable using [^18^F]FFNP in a preclinical study by Salem et al. In the same study, although the tracer could not differentiate between PRα and PRβ isoforms, it significantly distinguished PR-positive from PR-negative tumors [117].

ESR1 mutations can lead to resistance to hormone therapy. A study by Kumar et al. in preclinical models of PR-positive breast cancer xenografts, with and without ESR1 mutations, demonstrated that [^18^F]FFNP could effectively detect a decrease in tracer uptake in wild-type tumors after 7 days of endocrine therapy, while no decrease was observed in ESR1-mutant tumors, indicating therapy resistance [120]. These findings suggest that [^18^F]FFNP may aid in the early detection of therapy resistance induced by ESR1 mutations. Furthermore, [^18^F]FFNP was evaluated clinically by Dehdashti et al. for tracking PR changes in response to estradiol challenges in postmenopausal patients with advanced ER-positive breast cancer. The study found that post-challenge increases in tumor uptake could identify responders to endocrine therapy that the baseline scan could not. Patients with an increase in SUV_max_ > 7 were associated with significantly longer survival [121].

Several other promising candidates for progesterone receptor (PR) PET tracers are worth noting. In mouse models, Wu et al. showed [^18^F]EAEF could differentiate between PR-positive and PR-negative breast cancer tumors with high specificity (confirmed through blocking studies in vivo). However, its biodistribution shows significant accumulation in adipose tissue due to its lipophilicity, along with high uptake in the liver and gallbladder [122]. Another preclinical study by Lee et al. explored [^18^F]FPTP, a non-steroidal compound resistant to dehydrogenase activity, resulting in decreased liver uptake. Moreover, [^18^F]FPTP offers several advantages over [^18^F]FFNP, including lower bone uptake, suggesting better in vivo stability. Unlike [^18^F]FFNP, which also binds to the glucocorticoid receptor, limiting its specificity, [^18^F]FPTP is more selective. However, the production of [^18^F]FPTP is limited by the presence of enantiomeric byproducts, affecting its purity [123]. Another promising candidate, [^18^F]FPTT, demonstrated good specificity in a preclinical study by Gao et al., showing significantly higher uptake in PR-positive tumors compared to PR-negative ones, with its specificity validated by a blocking study in vivo and an increased tumor uptake following estradiol stimulation. The results highlight the tracer’s improved hydrophilicity, good stability, and moderate affinity [124]. Allott et al. demonstrated that a benzoxazinthione derivative of tanaproget labeled with ^18^F shows promising specificity in non-tumor-bearing preclinical models. However, it faces challenges due to rapid in vivo defluorination, requiring further optimization to improve stability [125].

### 2.4. Molecular Imaging of Ki67 Protein

Ki67 protein is active during the G1, S, G2, and M phases of the cell cycle but is not expressed in G0, anaphase, or telophase [126,127]. This makes Ki67 a key proliferation marker, particularly useful for differentiating HER2-negative luminal subtypes [3]. In the breast cancer population, the Ki67 index is approximately 25% [128], with higher levels generally associated with poor prognosis, higher metastasis rates, and increased recurrence [127,129]. High Ki67 expression varies across subtypes: it is lowest in Luminal A, followed by HER2 and TNBC, and is most frequent in Luminal B [129]. Assessing Ki67 in vivo can aid in more precise classification of breast cancer patients for optimal therapy, as Ki67 levels can predict response to systemic chemotherapy and their changes can be used to monitor therapy response [127].

Direct tracers targeting Ki-67 are not yet available. The most commonly used tracers to investigate proliferation are [^18^F]FDG and [^18^F]FLT. Some studies have shown a moderate correlation between FDG uptake and Ki-67 expression in lymphoma [130] and meningioma [131]. A meta-analysis by Deng et al. reported an overall correlation coefficient of *r* = 0.44 across various cancers and a similar correlation (*r* = 0.44) specifically in breast cancer [132].

Mixed results were seen when investigating the correlation between FLT and ki67 in various types of cancer (glioma, head and neck squamous carcinoma, and colorectal cancer) [133,134,135]. However, a meta-analysis by Chalkidou et al. revealed a strong overall correlation between FLT uptake and Ki-67 (*r* = 0.7), with particularly strong correlations observed in lung, brain, and breast cancers [136]. Further analysis by Surov et al. comparing these two tracers specifically in breast cancer found that FLT had a higher correlation with Ki-67 (*r* = 0.54) compared to FDG (*r* = 0.4) [137]. Based on this research, FLT appears to be a more relevant tracer for indirectly assessing Ki-67 levels. This is reasonable, as FLT reflects the activity of thymidine kinase, a crucial enzyme in the proliferation process [138].

A novel tracer, [^18^F]ISO-1, has been developed by McDonald et al. as a proliferation marker targeting the σ2 receptor. The study using this tracer found a moderate correlation between [^18^F]ISO-1 SUV_max_ and Ki-67 levels (*r* = 0.46) in invasive breast cancer patients [139].
medicina-60-02099-t001_Table 1Table 1Summary of radiopharmaceuticals assessing conventional biomarkers in breast cancer across preclinical and clinical studies.Imaging BiomarkerRadiopharmaceuticalsClinical/Preclinical PhaseKey FeaturesReferencesHER2Trastuzumab labeled ^89^Zr[^89^Zr]Zr-DFO-trastuzumabClinicalEffective in detecting HER2+ metastases, including brain lesionsHelped predict and follow up therapy response, especially when combined with FDG PET/CTChallenges include false positives in liver and bone possibly due to instability[29,30,31,32,33,34,35,36,37,38][^89^Zr]Zr-HOPO-trastuzumabPreclinicalImproved stability but lower tumor uptake compared to [^89^Zr]Zr-DFO-trastuzumab[39][^89^Zr]Zr-DFO*-trastuzumabPreclinicalImproved stability, retained high tumor contrast compared to [^89^Zr]Zr-DFO-trastuzumab, and was able to visualize HER2+ metastasis in bone[40][^89^Zr]Zr-DFO-^ss^trastuzumab-EndoS,[^89^Zr]Zr-DFO-^nss^trastuzumab-PNGaseFPreclinicalHigher tumor uptake and higher tumor contrast (especially against liver and spleen) compared to [^89^Zr]Zr-DFO-trastuzumabRequired enzymatic deglycolysating step[41]Trastuzumab labeled ^64^Cu[^64^Cu]Cu-DOTA-trastuzumabClinicalSuccessful for imaging HER2+ metastasis, including brain metastasesPotential in predicting therapy responsesNoted transchelation with serum compounds[43,44,45,46][^64^Cu]Cu-NOTA-trastuzumabClinicalHigher tumor contrast and lower off-target uptake and absorbed compared to [^64^Cu]Cu-DOTA-trastuzumab[47,48]Trastuzumab labeled ^52^Mn[^52^Mn]Mn-BPPA-trastuzumabPreclinicalHigher HER2+ tumor contrast than control [^52^Mn]Mn-DOTAGA-trastuzumab, allowed earlier differentiating of tumor HER2 positivityNeeded stability improvement[51][^52^Mn]Mn-Oxo-DO3A-trastuzumabPreclinicalComparable tumor contrast to [^52^Mn]Mn-BPPA-trastuzumab and good stability[52]Pertuzumab labeled ^89^Zr[^89^Zr]Zr-DFO-pertuzumabClinicalWas able to detect HER2+ metastasis, including brain metastasisShowed potential in predicting and follow-up therapy responses (even during trastuzumab treatment)Has slightly higher absorbed dose compared to [^89^Zr]Zr-DFO-trastuzumab[53,54,55,56,57][^89^Zr]Zr-^ss^DFO-pertuzumabClinicalImproved tumor-to-liver and -spleen ratios, but increased renal absorbed dose compared to [^89^Zr]Zr-DFO-pertuzumab[58,59]Trastuzumab emtansine labeled ^89^Zr[^89^Zr]Zr-DFO-T-DM1PreclinicalHigher HER2+ tumor uptake than [^89^Zr]Zr-DFO-trastuzumabPotential in predicting T-DM1 responses[60,61]Labeled trastuzumab fragments[^68^Ga]Ga-NOTA-F(ab’)-trastuzumab, [^68^Ga]Ga-NOTA-F(ab’)_2_-trastuzumabPreclinicalFaster clearance than trastuzumab while retain affinityF(ab’) tracer showed faster clearance and higher tumor-to-background ratios than F(ab’)_2_, except for tumor-to-kidney ratios[62][^64^Cu]Cu-MANOTA-Fab-trastuzumabPreclinicalBetter stability and tumor contrast than using other chelators (DOTA, DOTAGA, NODAGA) Clear visualization of the HER2+ tumor from 4h p.i.[63][^68^Ga]Ga-DFO-M74 trastuzumab’s FabPreclinicalBetter stability and higher tumor contrast compared to non-specific wild type Fab tracer[64][^89^Zr]∙DFO-HER2-Fab-PAS_200_ClinicalExtended biological half-life and improved tumor visualization including metastasis[65,66]HER2 nanobody[^18^F]TFPFN-5F7, [^18^F]TFPFN-2Rs15dPreclinicalLower kidney uptake, improved tumor contrast compared to tracers using other ^18^F prosthetics[67][^68^Ga]Ga-NOTA-2Rs15dClinicalConfirmed safety and reproducibility, able to detect inter-lesion heterogeneityDiscordance with IHC results posed challenges[68,69][^89^Zr]Zr-DFO-2Rs15dPreclinicalExplored the biodistribution of the tracer at later time points, kidney uptake remained a dosimetry challenge[70]HER2 affibody[^68^Ga]Ga-ABY-025ClinicalDetected HER2 heterogeneity and was able to follow HER2 targeting therapyTumor uptake correlation with IHC was not significant[74,75,76,77,78][^18^F]F-Al-MAL-NOTA-Cys-MZHER_2:342_PreclinicalGood HER2 specificity with lower liver uptake than ZHER_2:342_ tracers. Low bone uptake indicated tracer stability [79][^68^Ga]Ga-MAL-NOTA-Cys-MZHER_2:342_ClinicalAble to differentiate HER2 positivityLower liver uptake compared to ABY-025 tracer[80][^89^Zr]Zr-DFO-MAL-NOTA-Cys-MZHER_2:342_PreclinicalGood HER2 specificity with low liver and bone uptake; indirectly indicated tracer’s stability [81]Other HER2 scaffolds and peptides[^89^Zr]Zr-DFO-G3-DARPinPreclinicalHigh HER2 affinity and specificity but prominent kidney uptake and slow site-specific conjugation step[85][^68^Ga]Ga-DOTA-(Ser)_3_-LTVSPWYPreclinicalGood HER2+ tumor visualization, and tracer’s stability despite prominent kidney uptake[89][^68^Ga]Ga-DOTA-PEG_2_-DTFPYLGWWNPNEYRY, [^68^Ga]Ga-DOTA-PEG_2_-GSGKCCYSLPreclinical[^68^Ga]Ga-DOTA-PEG_2_-DTFPYLGWWNPNEYRY showed better human serum stability and more significant differentiation of HER2-positivty 1h p.i. than [^68^Ga]Ga-DOTA-PEG_2_-GSGKCCYSL[90]Estrogen Receptor Alpha
[^18^F]FESClinicalImproved metastasis detection in ER-positive patients when combined with FDG, especially in bone lesions Could predict and follow ER changes during endocrine therapy[97,98,99,100,101,102,103,104,105,106][^18^F]4FMFESClinicalImproved tumor contrast and lower organ uptake compared to [^18^F]FES, but high uterus and ovary uptake in preclinical models[107,109]Progesterone Receptor
[^18^F]FFNPClinicalDifferentiated PR-positive tumors; predicted hormone therapy responses and early detection of both PR and ER changes during the therapyChallenges include high uptake in liver and gastrointestinal tract[117,118,119,120,121][^18^F]EAEFPreclinicalDifferentiated tumor PR positivity, high adipose and abdominal uptake due to lipophilicity[122][^18^F]FPTPPreclinicalGreater PR selectivity and lower liver uptake compared to [^18^F]FFNP; reduced bone uptake indicated in vivo stabilityEnantiomeric byproducts may impair purity[123][^18^F]FPTTPreclinicalImproved hydrophilicity and stability; effectively distinguishes PR-positive tumors with significant specificity[124]Ki67
[^18^F]FDGClinicalCorrelation with Ki67 expression in breast cancer is moderate (r = 0.4)[132,137][^18^F]FLTClinicalStronger correlation with Ki67 in breast cancer (r = 0.54)[136,137][^18^F]ISO-1ClinicalModerate correlation with Ki67 in invasive breast cancer (r = 0.46)[139]


## 3. Novel Imaging Biomarkers

### 3.1. Epidermal Growth Factor Receptor Imaging

EGFR belongs to a family of transmembrane glycoproteins with similar structures and signaling pathways, whose interactions significantly influence cancer cell activity [140]. EGFR overexpression is observed in at least 15% of breast cancers [141]. Although expressed across all subtypes, EGFR is most commonly found in aggressive forms, particularly TNBC (around half of the cases) [140,142]. High EGFR expression generally correlates with poor outcomes, especially in TNBC, and is associated with higher metastasis rates. In contrast, in luminal subtypes, EGFR expression has been linked to improved survival [142]. Among HER2-enriched breast cancer population, the close interplay between EGFR and HER2 receptors also contributes to poor prognosis, as patients with co-expression of these proteins often exhibit therapy resistance [143]. Therefore, proper EGFR imaging can be a promising tool for selecting candidates for EGFR targeting therapy, potentially improving prognosis, especially in aggressive subtypes.

A study by Sadri et al. using [^64^Cu]Cu-DOTA-cetuximab on breast cancer xenografts with high levels of EGFR demonstrated efficient tumor specificity (confirmed with a blocking study in vivo), achieving a tumor-to-blood ratio of approximately 2 (2 h post-injection). Notably, after extended blocking (24 h blocking) with unlabeled cetuximab, the tumor exhibited increased tracer uptake, potentially due to the regeneration of EGFRs on the cell surface [144]. With the same antibody, a study by McKnight et al. using [^89^Zr]Zr-DFO-cetuximab on TNBC breast cancer xenografts with varying EGFR levels, the tracer effectively stratified EGFR expression. The tracer also showcased its potential in tracking dynamic changes in EGFR expression. Specifically, the increase in membranous EGFR under dasatinib treatment, which indirectly indicates the regaining of sensitization to cetuximab, strongly correlated (*r* > 0.8) with the increased tumor uptakes [145].

Similarly, Bhattacharyya et al., using [^89^Zr]Zr-DFO-panitumumab, demonstrated a strong correlation (*r*^2^ = 0.857) between tumor uptake and EGFR expression in mice carrying breast cancer xenografts, with notably elevated lymph node activity [146]. In a preclinical study by Cavaliere et al. on TNBC xenografts, [^89^Zr]Zr-DFO-amivantamab (a bispecific antibody against EGFR and cMET) demonstrated dual specificity, highlighting its potential for selecting candidates for amivantamab therapy [147].

Another study by Tikum et al., in preclinical models carrying colorectal and breast cancer, used [^89^Zr]Zr-DFO-matuzumab, targeting a different EGFR epitope compared to nimotuzumab, and showed high specificity for EGFR. The tracer’s uptake was significantly blocked by unlabeled matuzumab but not by nimotuzumab. Additionally, the tracer successfully visualized tumors regardless of KRAS mutation status, a factor known to induce anti-EGFR resistance and that can decrease tumor uptake [148]. In a separate study by Chekol et al., [^89^Zr]Zr-DFO-nimotuzumab effectively differentiated EGFR-positive from EGFR-negative breast cancer xenografts starting at 24 h post-injection, with the greatest distinction observed at 7 days post-injection. This tracer also demonstrated a lower absorbed dose compared to [^89^Zr]Zr-DFO-trastuzumab [149].

Further research by Solomon et al. used site-specific labeling with SpyTag/SpyCatcher to produce [^89^Zr]Zr-DFO-nimotuzumab- SpyTag-∆N-SpyCatcher. The tracer could visually differentiate EGFR-positive and -negative tumors in preclinical models, with high specificity for EGFR (a marked decrease in positive tumor uptake when imaging with [^89^Zr]Zr-control-IgG-SpyTag-∆N-SpyCatcher). Additionally, [^255^Ac]Ac-nimotuzumab labeled using this method showed higher survival rates compared to the unlabeled antibody, suggesting its potential for radioimmunotherapy [150].

Another preclinical study in breast cancer, colorectal, and melanoma tumors by Alizadeh et al. used an EGFR domain II-specific antibody fragment, 8709, labeled with ^89^Zr. This fragment, targeting a different domain than nimotuzumab, demonstrated good specificity. Tumor uptake was significantly blocked by unlabeled scFc-Fc, but not by nimotuzumab, and could distinguish EGFR positivity as early as day 2. However, as expected with small antibody fragments, kidney uptake remained the highest [151]. Recently, an EGFR affibody labeled with ^89^Zr using a novel cyclic fusarinine C chelator ([^89^Zr]Zr-FSC-ZEGFR:2377) was studied by Summer et al. in mice with epidermoid carcinoma xenografts, yielding favorable results both in vitro and in vivo [152].

### 3.2. Human Epidermal Growth Factor Receptor 3 Imaging

Human epidermal growth factor receptor 3 (HER3), also a member of the EGFR/ERBB family of receptor tyrosine kinases, plays a crucial role in cancer progression by activating the PI3K-AKT-mTOR pathway upon heterodimerization with other receptors [153]. Among the breast cancer population, HER3 overexpression was found in at least more than 18% of the cases [154]. It is especially associated with the HER2 subtype, showing a tendency for brain metastasis, which reduces survival rates and contributes to resistance against HER2-targeted therapies [155]. Additionally, HER3 overexpression worsens prognosis in TNBC, particularly when co-expressed with EGFR [156]. As a result, HER3-targeting therapies, such as lumretuzumab and patritumab, have been actively studied, showing promising efficacy and tolerable side effects [154]. Early in vivo HER3 imaging may enhance the prediction of response to HER3-targeted therapies in breast cancer, supporting more personalized treatment approaches.

Several studies have investigated the use of [^89^Zr]-labeled antibodies for imaging breast cancer in preclinical and clinical settings. Bensch et al. utilized [^89^Zr]Zr-DFO-lumretuzumab to target HER3 in patients with advanced cancers, including breast cancer. It was found that the optimal tumor-to-background contrast was achieved with a dose of 100 mg of unlabeled lumretuzumab and with imaging performed 4 or 7 days post-injection. This approach successfully visualized 67.6% of lesions ≥ 10 mm detected by CT, including some brain lesions and previously undetected bone metastases. However, the detection of lung and liver lesions was less reliable, and uptake did not correlate with HER3 expression as determined by IHC [157]. In a separate clinical study (n = 6) by Oordt et al. using [^89^Zr]Zr-DFO-GSK2849330, another HER3-targeting antibody, it was observed that an 8 mg dose provided the best tumor contrast on day 5 post-injection. This study also noted that tumor uptake was saturable at a dose of 30 mg/kg, demonstrating the tracer’s specificity for targeting HER3. The tracer successfully visualized and quantified lesions in bones, soft tissues, and lungs. Nonetheless, there was no correlation between lesion uptake and the treatment response to the antibody [158].

Imaging with HER3 antibodies to assess therapy effects has been conducted in breast cancer-bearing mice. In one study by Pool et al., [^89^Zr]Zr-DFO-mAb3481 was used during lapatinib therapy. This tracer achieved tumor-to-blood ratios exceeding 50—higher than those observed in other studies using HER3 affibodies, which have the advantages of high affinity and rapid clearance for high tumor contrast. The study found no changes in tumor uptake, which was consistent with the stable HER3 expression in xenografts during treatment as confirmed by IHC, thus validating the tracer’s accurate HER3 expression assessment [159].

Antibody fragments targeting HER3 show great promise for assessing HER3 status in breast cancer preclinical models. For instance, [^64^Cu]Cu-CB-TE2A-F(ab’)2-mAb105 was used by Wehrenberg-Klee et al. to monitor cellular changes under AKT and PI3K inhibitors, demonstrating a significant relationship between tumor uptake and increased HER3 levels after therapy. This tracer also effectively detected the treatment-induced increase in receptor tyrosine kinase levels [160].

HER3 affibodies offer a significantly higher tumor contrast than antibodies or antibody fragments, as shown by compaing ZHER_3:08698_ with seribantumab and its fragments [161]. A study by Rosestedt et al. using [^68^Ga]Ga-HEHEHE-ZHER_3:8698_-NOTA in preclinical models with various cancers, including breast cancer, demonstrated the tracer’s high specificity for HER3, showing a significant correlation (*r* = 0.66) between tumor uptake and HER3 expression. The tracer exhibited significantly higher uptake in HER3-positive breast cancer xenografts compared to HER3-negative tumors and could become saturated with an excess dose of the affibody [162]. Another study by Da Pieve et al. compared [^18^F]AlF-NOTA-ZHER_3:8698_ with [^18^F]AlF-NODA-ZHER_3:8698_ in breast cancer mouse models, revealing that both tracers visualized the tumors 1 h post-injection. However, the NOTA-conjugated tracer provided better TBRs, though its chelation process requires high temperatures, which can be challenging for heat-sensitive peptides [163]. HER3 affibody tracers can also be used to evaluate therapy resistance, as the activation of the HER3 pathway is one of the mechanisms responsible for resistance to anti-HER2 therapy. In a preclinical model by Martins et al., the HER3-specific tracer [^89^Zr]Zr-DFO-ZHER_3:8698_ demonstrated high specificity for HER3, effectively stratifying HER3 expression in correlation with IHC staining. This tracer was used to assess changes in HER3 uptake following HSP90 inhibitor treatment. Two weeks after initiating treatment, tumor uptake of the tracer significantly increased, corresponding to elevated HER3 expression [164]. In another preclinical study by Wehrenberg-Klee et al., the peptide-based tracer [^68^Ga]Ga-NOTA-ßAGGG-CLPTKFRSC effectively detected changes in HER3 expression (with nearly a threefold increase in uptake) in HER2-positive lapatinib-resistant xenografts after 2 days of treatment, enabling the timely selection of a more effective treatment plan by supplementing HER3-targeting siRNA along with lapatinib [165].

### 3.3. Vascular Endothelial Growth Factor Imaging

According to IHC results, at least 70% of breast cancer cases are positive for VEGF [166]. VEGF expression is most frequently associated with HER2-positive subtypes, followed by Luminal B and TNBC, and is least common in Luminal A subtypes [166,167]. VEGF, particularly VEGF-A, is crucial for the angiogenesis required for tumor growth, with VEGF receptors such as VEGFR1 and VEGFR2, which have a high affinity for VEGF-A, playing vital roles in breast cancer progression [168]. While recent therapies targeting VEGF-A or VEGFR2 [169,170] have shown mixed results, new anti-angiogenic agents are currently being explored in breast cancer treatment [171]. Thus, assessing VEGF and VEGFR expression in vivo may aid in selecting patients who are suitable candidates for these therapies.

Imaging VEGF with [^89^Zr]Zr-DFO-bevacizumab has been studied in breast cancer patients by Gaykema et al., showing clear tumor visualization with good contrast 4 days post-injection. Tumor uptake was significantly higher than for normal breast tissue, and SUV_max_ correlated with VEGF IHC (*r* = 0.49). Interestingly, increased uptake in the nipples was noted, likely due to elevated perfusion in these areas [172]. In a preclinical TNBC xenograft study by Scheltinga et al., the same tracer visualized a decrease in VEGF expression following HSP90 inhibitor treatment compared to the baseline uptake [173]. However, a clinical study by Gaykema et al. demonstrated that only the tumor uptake of [^89^Zr]Zr-DFO-trastuzumab, and not of [^89^Zr]Zr-DFO-bevacizumab, correlated with tumor size changes following HSP90 inhibitor treatment, as assessed by CT imaging [37].

A recent preclinical study by Yang et al. used [^89^Zr]Zr-DFO-aflibercept (Abe), a fusion protein targeting VEGF-A, VEGF-B, and placental growth factor (PlGF), to visualize TNBC xenografts. The tracer demonstrated high tumor contrast with confirmed specificity; however, it also showed high uptake in the liver, identifying the liver as the critical organ [174].

VEGF peptides labeled with PET isotopes have also been used for imaging VEGFR in TNBC breast tumor-bearing mice, especially VEGFR2, which is the crucial target in antiangiogenic therapy. In contrast, VEGFR1 is predominantly expressed physiologically in the kidneys. Wang et al. developed a mutated recombinant form of VEGF_121_, known as VEGF_DEE_, labeled with ^64^Cu via a DOTA chelator to selectively target VEGFR2. This tracer demonstrated slightly higher tumor uptake and reduced kidney uptake at all time points compared to [^64^Cu]Cu-DOTA-VEGF_121_. Nevertheless, both tracers showed strong tumor specificity, with uptake significantly blocked by the addition of unlabeled peptides in vivo [175]. Another tracer developed by Zhang et al., [^61^Cu]Cu-NOTA-K3-VEGF_121_, which features 3 lysine modification at the N-terminus for easier labeling, also demonstrated good specificity in a blocking study in vivo using an unlabeled compound, although the liver, gastrointestinal tract, and kidney showed remarkable activity [176]. In another study by Meyer et al., [^89^Zr]Zr-DFO-labeled single-chain VEGF-A mutants targeting VEGFR1 and VEGFR2 were compared with pan-receptor-selective scVEGF in TNBC-bearing mice. All compounds successfully visualized tumors at 2 h post-injection, with specificity validated by blocking studies in vivo with corresponding unlabeled peptides. However, kidney uptake was prominent, particularly with scVEGF-R1, due to physiological VEGFR expression [177].

In addition to renal VEGFR1 expression, many conditions involving neovascularization, such as atherosclerosis, stroke, and osteoarthritis, can provide a false positive when imaging tumor VEGFR, especially in the brain and bones, which are frequent sites of breast cancer metastasis. In fact, molecular imaging targeting these receptors has been used to visualize plaque [178] and post-stroke neoangiogenesis [179].

### 3.4. Androgen Receptor Imaging

The androgen receptor (AR) is expressed in over 70% of breast cancer cases, with positivity most prevalent in luminal subtypes, followed by TNBC [180]. In luminal subtypes, AR positivity has a notable impact on prognosis, as patients who are positive for both androgen and estrogen receptors generally experience better outcomes than those who are AR-negative [181]. Furthermore, in the TNBC subtype, AR positivity is observed in more than half of cases [182] and is associated with improved prognosis, suggesting that these patients may benefit from anti-androgen therapies [182,183]. Given the significant role of AR in breast cancer prognosis and treatment response, effective imaging to assess AR expression could greatly enhance patient selection for anti-androgen therapies, leading to more personalized and potentially more effective treatment strategies.

[^18^F]FDHT has been employed for androgen receptor (AR) imaging, with a study by Venema et al. in postmenopausal hormone receptor-positive breast cancer patients showing the tracer’s ability to visualize tumors with a sensitivity of 91% and a specificity of 100%. The tracer’s SUV_max_ was found to correlate moderately (*r*^2^ = 0.47) with AR IHC [184]. In a longitudinal study by Boers et al. in postmenopausal patients with AR-positive metastatic breast cancer receiving bicalutamide therapy, [^18^F]FDHT was able to track reductions in tumor uptake 4–6 weeks following treatment, but these reductions did not correlate consistently with therapy responses. Notably, significantly greater reductions were observed in ER-negative patients [185]. Another study by Jacene et al. involving serial imaging in estrogen receptor-positive postmenopausal patients undergoing selective AR modulation (SARM) therapy with GTx-024 demonstrated a greater reduction in tumor uptake in patients experiencing clinical benefit, although the difference was not statistically significant and the sample size was limited (n = 11) [186]. Recently, [^18^F]enzalutamide was evaluated in prostate cancer xenografts, demonstrating high specificity and stability as an AR-targeted tracer, suggesting that [^18^F]enzalutamide could potentially be adapted for imaging androgen receptors in breast cancer [187].

### 3.5. Estrogen Receptor Beta Imaging

Estrogen receptor beta (ERβ) is physiologically expressed in various organs, including the ovaries, central nervous system, male reproductive organs, colon, kidneys, and immune system [188]. Estrogen beta plays a protective role in breast cancer. Its expression levels are significantly lower in breast cancer tissues compared to healthy breast tissue, especially in higher-grade tumors [189]. ERβ positivity is observed in over 68% of breast cancer cases, with the highest prevalence in luminal subtypes [190], where it is generally associated with a favorable prognosis [191]. In TNBC, ERβ expression appears to be linked to better outcomes and treatment efficacy through multiple pathways [192]. However, the prognostic significance of ERβ in TNBC remains controversial due to the presence of various receptor isoforms, which may contribute to the complexity of its role in this subtype [193]. Given its protective role and prognostic value, ERβ-targeted imaging could be crucial for predicting the effectiveness of ERβ agonists, aiding in patient selection and personalized treatment.

In efforts to develop an ERβ tracer, a study using [^18^F]FEDPN in ERα-knockout and ERβ-knockout mice demonstrated ovary and uterus uptake mediated by ERβ. However, the tracer’s specificity was moderate, as the ovary—despite being the organ with the highest ERβ expression—showed only minor uptake differences between the knockout models [194].

Other studies on ovarian and prostate xenografts in preclinical models used [^18^F]PVBO, which confirmed specificity through blocking studies [195], and [^18^F]FHNP, where tumor uptake correlated with ERβ expression, validating its receptor selectivity [196]. Although no studies have yet been conducted in breast cancer-bearing mice, the tracers mentioned above show significant potential for assessing ERβ expression in breast cancer. Nevertheless, caution is required as there is considerable physiological expression of the receptor in organs such as the ovary and uterus.

### 3.6. Poly (Adenosine Diphosphate [ADP]-Ribose) Polymerase 1 Imaging

Poly (ADP-ribose) polymerase 1 (PARP1) is a crucial enzyme in the DNA repair pathway [197], making it a significant target for synthetic lethality in cancers with BRCA1 and BRCA2 mutations. PARP inhibitors have shown promise in improving survival rates for patients with these mutations [198]. Particularly, high PARP1 expression is generally associated with a poorer prognosis [199]. Although BRCA mutations often correlate with elevated PARP1 levels [199], not all BRCA-mutated cancers respond effectively to PARP inhibitors, and some non-BRCA-mutated cancers also exhibit high PARP1 expression [200]. Furthermore, PARP1 expression varies across breast cancer subtypes: HER2-positive and TNBC show higher nuclear expression, whereas luminal subtypes display increased cytoplasmic and overall expression [201,202]. The complex heterogeneity of breast cancer emphasizes the importance of assessing PARP1 expression in vivo to optimize the personalized and effective use of PARP1 inhibitors in therapy.

[^18^F]FluorThanatrace ([^18^F]FTT) has been used to assess PARP1 expression in preclinical studies. Zhou et al. demonstrated its specificity by visualizing TNBC tumors at 60 min post-injection, with uptake significantly decreased by an unlabeled compound or olaparib [203]. Tumor uptake correlated with PARP1 expression levels [204], and despite high abdominal uptake due to hepatobiliary clearance [203], tumor contrast was sufficient, with a tumor/muscle ratio of 1.9 observed in the group with the highest PARP1 expression [204]. In a small study by McDonald et al. (n = 4) in breast cancer patients, the tracer detected a reduction in SUV_max_ post-PARPi therapy, highlighting its potential for monitoring PARP1 expression in clinical settings [205].

Another candidate studied by Xu et al., [^18^F]-PARPi, an [^18^F]-labeled olaparib derivative, showed high specificity in TNBC xenografts, with tumors visualized 1 h post-injection, and blocking studies with olaparib reduced tumor uptake. Additionally, the block study also decreased spleen and pancreas uptake, explaining the elevated abdominal uptake observed in PARP1 imaging studies [206]. In a comparative study by Stotz et al. in breast cancer-bearing mice, their novel [^18^F]FPyPARP showed the highest tumor-to-blood ratios, while [^18^F]-PARPi had the highest tumor-to-muscle and tumor-to-kidney contrast, and [^18^F]FTT exhibited the highest tumor uptake and longer retention [207].

[^18^F]Talazoparib, the most potent PARPi labeled with ^18^F, was evaluated by Bowden et al. in breast cancer models, showing high specificity and significant tumor blocking by talazoparib and olaparib in vitro. The tracer also demonstrated exceptionally high tumor-to-blood ratios (around 10), outperforming [^18^F]FTT and [^18^F]-PARPi, with similar hepatic and renal clearance, though its enantiomer complicates tracer production [208].

Other potential PARP1-targeting tracers have been tested in preclinical models carrying breast cancer. Shuhendler et al. used [^18^F]SuPAR to monitor PARP1 levels with high specificity, evidenced by a significant reduction in tumor uptake when treated with talazoparib. There were increases in tumor uptake in a dose- and time-dependent manner following radiotherapy, which activates PARP1 in the similar manner [209]. Zheng et al. developed [^18^F]BIBD-300, a tracer with reduced lipophilicity favoring renal clearance, and evaluated it in breast cancer-bearing mice. Although [^18^F]BIBD-300 showed lower tumor uptake compared to [^18^F]FTT, it provided superior tumor contrast, with significantly higher tumor-to-muscle and tumor-to-liver ratios [210].

### 3.7. Programmed Death-Ligand 1 Imaging

Most breast cancers have some degree of tumor-infiltrating lymphocytes (TILs), with approximately 10% showing more than 50% lymphocytic infiltration. This group of predominant TIL tumors is generally associated with better outcomes in breast cancer patients and often shows elevated expression of PD-L1 [211]. Notably, there is a correlation between PR ER negativity and high PD-L1 expression. Consequently, HER2-enriched and TNBC subtypes both show particularly high PD-L1 levels, particularly in TNBC, despite these subtypes’ aggressive nature [212]. Given this, immune checkpoint inhibitor therapy in breast cancer can be more beneficial in treating TNBC [211]. However, the response rate in metastatic TNBC has been moderate, with a PFS of 2 months and an overall survival (OS) of 8.8 months, even in the PD-L1-positive patients [213]. The prognosis can be improved when combined with chemotherapy, especially in cancer with high PD-L1 expression, i.e., the PFS for the combined group was significantly longer than for the chemotherapy group, (with a hazard ratio of 0.69, even though the OS was not significantly different) [214]. Therefore, a noninvasive, in vivo assessment of PD-L1 status could be a valuable tool for selecting patients who would benefit from immune checkpoint inhibitor therapy.

In preclinical TNBC models, Massicano et al. utilized [^89^Zr]Zr-DFO-atezolizumab to image PD-L1 expression. The tracer effectively visualized tumors 7 days post-injection, with specificity confirmed through both in vitro and in vivo blocking studies. However, substantial uptake in the spleen, liver, kidneys, and adrenal glands led to high absorbed doses in these organs and limited tumor contrast. The study further demonstrated elevated tracer tumor contrast following treatment with niraparib (a PARP inhibitor) or chemotherapy; both upregulate PD-L1 expression [215]. In a clinical study by Bensch et al. involving patients with various cancer types, including TNBC, imaging performed at 4 and 7 days post-injection (preceded with 10 mg of unlabeled atezolizumab) revealed significant tracer accumulation in the liver, spleen, bone marrow, and lymphoid tissues. Notably, higher tumor SUV_max_ on PET images was more strongly correlated with positive responses to PD-L1 therapy than PD-L1 IHC results [216].

In preclinical TNBC models, Jagoda et al. demonstrated that [^89^Zr]Zr-DFO-avelumab exhibited a similar biodistribution, with high radioactivity levels in the spleen, lymph nodes, liver, bone marrow, and moderate uptake in tumors. Blocking studies using escalating doses (10 μg up to 400 μg) of unlabeled avelumab showed decreased spleen and liver uptake, with increased tumor and lymph node uptake at lower doses [217]. In a separate study by Li et al., pre-blocking with 1.5 mg of cold avelumab similarly reduced uptake in both the tumor and spleen, while liver uptake increased, suggesting strong non-tumor-specific binding in the spleen [218].
medicina-60-02099-t002_Table 2Table 2Summary of radiopharmaceuticals assessing novel biomarkers in breast cancer across preclinical and clinical studies.Imaging BiomarkerRadiopharmaceuticalsClinical/Preclinical PhaseKey FeaturesReferencesEGFRLabeled antibody[^64^Cu]Cu-DOTA-cetuximabPreclinicalEfficient tumor specificity in EGFR-expressing breast cancer; tracked the increase in EGFR after extended blocking[144][^89^Zr]Zr-DFO-cetuximabPreclinicalStratifies EGFR expression in TNBC xenografts; correlated with treatment-induced receptor changes[145][^89^Zr]Zr-DFO-panitumumabPreclinicalStrong correlation between uptake and EGFR expression; lymph node activity observed in xenografts[146][^89^Zr]Zr-DFO-amivantamabPreclinicalDemonstrated dual specificity for EGFR and cMET[147][^89^Zr]Zr-DFO-matuzumabPreclinicalHigh specificity for EGFR; visualized tumors regardless of KRAS mutation status[148][^89^Zr]Zr-DFO-nimotuzumabPreclinicalDifferentiates EGFR-positive/negative tumors; lower absorbed dose compared to [^89^Zr]Zr-DFO-trastuzumab[149][^89^Zr]Zr-DFO-nimotuzumab- SpyTag-∆N-SpyCatcherPreclinicalHigh specificity with reduced off-target effects; potential for theragnostic radioimmunotherapy.[150]Labeled antibody fragment[^89^Zr]Zr-DFO-8709-scFv-FcPreclinicalEarly to differentiate EGFR positivity (2 days p.i.) with high specificityKidney uptake was prominent[151,157]HER3Labeled HER3 antibody[^89^Zr]Zr-DFO-lumretuzumabClinicalVisualizes HER3-positive tumors, including brain metastases; limited reliability in lung/liver lesions[157][^89^Zr]Zr-DFO-GSK2849330ClinicalSaturable uptake; visualized lesions in bones, soft tissues, and lungs; no correlation with treatment response[158][^89^Zr]Zr-DFO-mAb3481PreclinicalHigher tumor-to-blood ratios overserved in HER3 affibodies, following HER3 expression under therapy[159]Labeled antibody fragments[^64^Cu]Cu-CB-TE2A-F(ab’)2-mAb105PreclinicalMonitors HER3 level and level of tyrosine kinase activity changes under therapy[160]HER3 affibody[^68^Ga]Ga-HEHEHE-ZHER_3:8698_-NOTAPreclinicalHigh specificity and tumor contrast with significant correlation between tumor uptake and HER3 level[162][^18^F]AlF-NOTA-ZHER_3:8698_PreclinicalPromising HER3 visualization as early as 1h p.i.; better tumor-to-background ratio than conjugation with NODA[163][^89^Zr]Zr-DFO-ZHER_3:8698_PreclinicalHigh specificity; effective for assessing HER3 expression changes under therapy[164]HER3 peptide[^68^Ga]Ga-NOTA-ßAGGG-CLPTKFRSCPreclinicalEffective for assessing HER3 expression changes under therapy[165]VEGF/VEGFRVEGF antibody[^89^Zr]Zr-DFO-bevacizumabClinicalVisualizes VEGF-positive tumors; correlates moderately with VEGF IHC; high uptake in nipples observedPotential to follow VEGF expression changes during therapy[37,172,173]VEGF antagonist[^89^Zr]Zr-DFO-afliberceptPreclinicalTargets VEGF-A/B and PlGF; high tumor contrast; liver identified as critical organ[174]VEGF and derivatives[^64^Cu]Cu-DOTA-VEGF_DEE_PreclinicalSelectively targets VEGFR2; strong tumor specificity with reduced kidney uptake compared to VEGF_121_ tracer[175][^61^Cu]Cu-NOTA-K3-VEGF_121_PreclinicalFacile labelling with good specificity but elevated liver, gastrointestinal tract, kidney uptakes[176]Single-Chain VEGF Mutants[^89^Zr]Zr-DFO-scVR1, [^89^Zr]Zr-DFO-scVR2PreclinicalHigh specificity with the respective receptors (VEGFR1 and VEGFR2); kidney uptake was prominent[177]Androgen Receptor
[^18^F]FDHTClinicalVisualizes AR-positive tumors; correlated moderately with AR IHC; tracks therapy-induced uptake changes[184,185,186]PARP1
[^18^F]FTTClinicalCorrelates with PARP1 expression; sufficient tumor contrast; high abdominal uptake noted in preclinical resultsPotential to track PARP1 reduction after therapy [203,204,205,207][^18^F]-PARPiPreclinicalHigh specificity in TNBC xenografts; reduced uptake in spleen and pancreas with blocking studies[206,207][^18^F]FPyPARPPreclinicalShowed the highest tumor-to-blood ratios compared to [^18^F]FTT and [^18^F]-PARPi[207][^18^F]TalazoparibPreclinicalHigher tumor-to-blood ratios than [^18^F]FPyPARP; challenging tracer production with enantiomer[208][^18^F]SuPARPreclinicalAble to track the PARP1 level changes after therapy in a dose- and time-dependent manner[209][^18^F]BIBD-300PreclinicalReduced lipoplicity to improve tumor-to-liver ratios [210]PD-L1PD-L1 antibody[^89^Zr]Zr-DFO-atezolizumabClinicalAble to predict and follow up therapy responsesHigh uptake in spleen, liver, and bone marrow limited tumor contrast[215,216][^89^Zr]Zr-DFO-avelumabPreclinicalEffective in imaging PD-L1; biodistribution varied with pre-blocking, high specific spleen uptake [217,218][^64^Cu]Cu-NOTA-MX001PreclinicalImproved tumor contrast that overcame spleen liver uptakes[219][^64^Cu]Cu-NOTA-durvalumabPreclinicalHigh specificity, further bettered tumor-to-liver and -spleen ratios but increased blood pool uptake[220][^89^Zr]Zr-p-SCN-Bn-C5HOPO-STM108PreclinicalSuperior tumor contrast compared to [^89^Zr]Zr-DFO-STM108, with tumor uptake higher than liver and spleen uptakes[221]


In another approach by Xu et al., [^64^Cu]Cu-NOTA-MX001, a fully human anti-PD-L1 antibody labeled with ^64^Cu was used in mice bearing PD-L1-positive colorectal cancer and -negative TNBC cancer. Despite initially elevated liver and spleen uptake, PD-L1-positive tumors became the highest uptake tissue by day 2 post-injection [219]. Similarly, [^64^Cu]Cu-NOTA-durvalumab was used by Malih et al. in TNBC and pancreatic cancer xenografts, demonstrating high specificity. PD-L1-positive TNBC tumors showed approximately fivefold higher uptake than negative tumors 2 days post-injection, while significantly lower uptake was observed in positive tumors using a [^64^Cu]Cu-NOTA-IgG control. Moreover, this tracer provided excellent tumor contrast, with the tumor exhibiting the highest uptake, followed by the blood, while liver and spleen uptake remained moderate [220]. A novel PD-L1 tracer, developed by Radaram et al., [^89^Zr]Zr-p-SCN-Bn-C5HOPO-STM108, was tested in PD-L1-positive TNBC xenografts and showed superior tumor contrast compared to [^89^Zr]Zr-DFO-STM108. Specifically, 6 days post-injection, the tumor exhibited the highest uptake, with minimal bone uptake, highlighting the tracer’s stability and its potential for PD-L1 imaging [221].

## 4. Future Directions

The future of PET imaging in breast cancer lies in the development of radiotracers that are more stable and highly specific, enhancing tumor contrast and targeting a broader range of biomarkers while minimizing off-target radiation. Efforts should focus on optimizing tracers distinguishing between established biomarkers, such as HER2 and ERα, and developing emerging targets, like ERβ and HER4, to better capture the molecular heterogeneity of breast cancer.

Addressing the challenges of radiotracer selection is critical. Antibody-based tracers are highly specific and valuable for evaluating antibody biodistribution and future radioimmunotherapy dosimetry, but they are hindered by long circulation times and limited tissue penetration. In contrast, small molecules, with their smaller size, faster clearance, better tissue penetration, and high binding affinity, provide a promising alternative despite issues like high kidney uptake. Isotope selection must also be addressed for labeling bioactive molecules: antibodies require long half-life isotopes such as ^89^Zr and ^52^Mn, while small molecules benefit from short half-life isotopes like ^18^F and ^68^Ga. Selected isotopes should possess low positron maximum energy and high frequency for optimal resolution, along with low-energy gamma emissions for radiation safety. Attention must also be given to prosthetic groups or chelators for better high labeling, conjugating efficiency, and stability, ensuring compatibility without compromising tracer bioactivity.

However, potential pitfalls such as off-target uptake must be addressed, as they contribute to false positives, increased radiation dose, and reduced tumor contrast. These include the following: tracer instability (e.g., free ^18^F or ^89^Zr in bone; free ^52^Mn in the pancreas); metabolic pathways (e.g., liver metabolism for antibody or lipophilic tracers like hormone receptor tracers; kidney excretion for small molecules); physiological expression of targets in organs (e.g., HER2 in the liver [222] and ovary [223]; hormone receptors in the uterus, ovary and pituitary gland [224]; PD-L1 in lymphoid tissues; and VEGFR1 in the kidneys); and non-malignant conditions increasing target expression (e.g., VEGFR in atherosclerosis, post-stroke neovascularization, and PARP1 in fatty liver disease [225]). Moreover, limited tumor expression of certain targets, such as HER3, highlights the need for tracers with high affinity to achieve sufficient imaging.

Future research should prioritize innovations in tracer development, including the optimization of suitable isotope production, facile site-specific labeling methods, and pharmacokinetic improvements to enhance safety (addressing both immunogenicity and dosimetry concerns) and effectiveness. Establishing standardized protocols is essential to minimize false negatives and false positives while ensuring accurate biomarker stratification, prediction, and the timely monitoring of therapy responses. These advancements will drive the evolution of PET imaging, paving the way for more precise and effective personalized breast cancer therapies.

## 5. Conclusions

In conclusion, the ongoing evolution of PET imaging and the development of innovative radiotracers hold significant promise for enhancing the diagnosis, treatment, and monitoring of breast cancer. By focusing on improving tracer specificity, stability, and safety, future research can pave the way for more personalized and effective management strategies, ultimately improving patient outcomes.

## Figures and Tables

**Figure 1 medicina-60-02099-f001:**
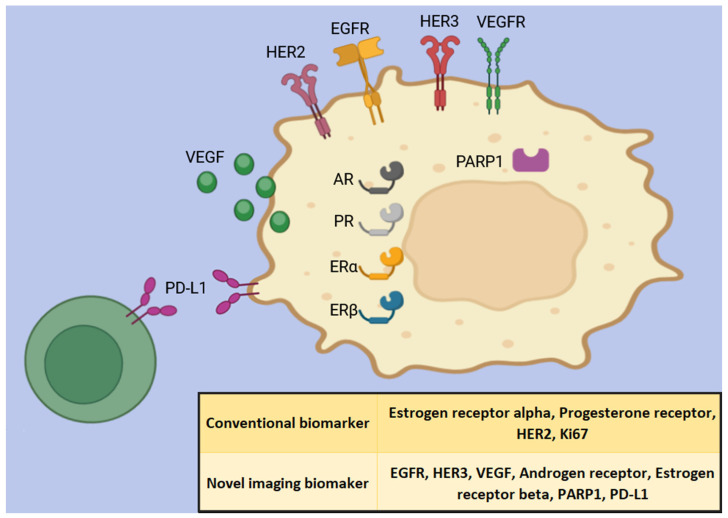
An overview of the imaging targets for breast imaging discussed in this review. The green, smaller cell in the bottom-left corner represents a white blood cell, while the larger, orange cell on the right represents a breast cancer cell. This figure was created with BioRender.com.

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
