# Peer review of "Novel Molecular Classification of Breast Cancer with PET Imaging"

_medicina, 2024, doi:10.3390/medicina60122099_

Round 1

Reviewer 1 Report

Comments and Suggestions for Authors

The review manuscript titled "Novel Molecular Classification of Breast Cancer with PET Imaging" focuses on both clinical and preclinical studies, exploring the latest advancements in PET imaging for breast cancer. It emphasizes the development of novel tracers targeting hormone receptors, including estrogen receptor alpha, progesterone receptor, androgen receptor, and estrogen receptor beta, as well as the ErbB receptor family, VEGF/VEGFR, PARP1, PD-L1, and markers for the indirect assessment of Ki-67. The manuscript is compelling, contains valuable information, and is well-researched. However, to enhance its clarity, structure, and overall impact, the following improvements are recommended to make it suitable for publication:

1. The current version of Figure 1 lacks sufficient detail. Please provide additional context and explicitly specify the cell types represented in the figure to improve its utility and clarity.

2. The specific aims of the review are currently limited to a single sentence at line 94. Expanding this section to clearly outline the scope and objectives of the manuscript will help readers better understand what the review intends to cover and what they can expect.

3. Heading 2 begins abruptly after the introduction. Adding a transitional section to establish context and connect the introduction to this section would enhance the manuscript's flow and readability.

4. Some headings, such as "Ki-67 Protein" at line 481, are too simplistic. Consider revising them to provide more descriptive and contextual titles that better reflect the content of each section.

5. Several sentences in the manuscript are excessively long, which may hinder reader comprehension. Rewriting these sentences to make the points clearer and more concise is recommended.

6. While this is at the authors’ discretion, it may be more appropriate to italicize the term in vivo throughout the manuscript for consistency with scientific convention.

Reviewer 2 Report

Comments and Suggestions for Authors

The manuscript provides a comprehensive review of molecular classification in breast cancer using PET imaging. It thoroughly examines the clinical and preclinical applications of various radiopharmaceuticals. However, transitions and content balance between subsections could be improved.

Comments:

1.       The abstract could better articulate the limitations of current methods and the necessity for new imaging approaches in a clearer and more concise manner.

2.       Rewrite the introduction section concisely, removing redundant or unnecessary details. Exploration of breast cancer risk factors is unnecessary in this context.

3.       Define abbreviations clearly, such as "RGD."

4.       Emphasize the need for imaging tailored to address tumor heterogeneity and specific subtypes.

5.       Highlight succinctly that tumor types may change their characteristics with metastasis, and the importance of novel radiopharmaceuticals for tracking these changes. Stress their role in understanding tumor behavior patterns.

6.       The discussion on HER2 radiopharmaceuticals is detailed, but clinical implications of these findings should be more explicitly stated.

7.       Provide a brief definition of "HER2-low" for clarity.

8.       The HER2 subsection lacks coherence and focus. Revise it to be more concise and structured, focusing on:

    1. Advantages and disadvantages of new methods.
    2. How these methods address limitations of current approaches.
    3. Any inherent limitations they themselves may possess.

9.       ER-Positive Patients:

The statement: "In ER-positive patients previously eligible for endocrine therapy, treatment with vorinostat (an HDAC inhibitor) combined with an aromatase inhibitor showed benefits in half of the patients, possibly due to vorinostat’s ability to restore ER sensitivity," needs clarification.

What is the definition of "benefit" here?

  1. Provide numeric data to support claims to enhance scientific rigor. Avoid overly general and superficial language, even in a narrative review.

11.   Table 1:

While informative, the table could be visually enhanced using color coding and by adding columns detailing the advantages, disadvantages, and limitations of each radiopharmaceutical.

12.   VEGFR Section:

Discuss how VEGFR imaging may be influenced by other conditions causing neovascularization in the body.

  1. Highlight potential pitfalls such as false negatives and false positives, along with key considerations and limitations for each imaging technique.

14.   PD-L1 Section:

The statement: "Given this, immune checkpoint inhibitor therapy in breast cancer is mostly implemented in treating TNBC. Although the response rate has been moderate, it can be improved when combined with chemotherapy, especially in PD-L1-positive cancer," requires more specific information:

Cite relevant studies.

Include progression-free survival (PFS) or overall survival (OS) statistics.

Specify if the data pertains to advanced or early-stage disease.

15.   Future Directions:

The discussion on the future potential of molecular imaging techniques should be more specific. Address:

Challenges ahead in the field.

What innovations or developments can be expected.

Propose hypotheses that could guide future studies, offering a blueprint for upcoming research.

16.   Revise long sentences into shorter, more straightforward ones to improve readability.

17.   The manuscript, while thorough, is difficult to read due to its density and complexity. A major revision is needed to make it more user-friendly. Current explanations are too general, lacking sufficient numerical data to support the claims.

18.   The manuscript’s strength lies in its focused effort to consolidate an extensive amount of information on a specific topic. However, significant revisions are required to enhance its readability and scientific depth.

Round 2

Reviewer 1 Report

Comments and Suggestions for Authors

In the updated manuscript “Novel Molecular Classification of Breast Cancer with PET Imaging” the authors have successfully addressed all previous concerns. The manuscript now convincingly presents innovative perspectives on latest advancements in PET imaging for breast cancer. Therefore, I recommend this article for publication.

Reviewer 2 Report

Comments and Suggestions for Authors

The authors have addressed all my concerns